# CRISPR/dCas9 Tools: Epigenetic Mechanism and Application in Gene Transcriptional Regulation

**DOI:** 10.3390/ijms241914865

**Published:** 2023-10-03

**Authors:** Ruijie Cai, Runyu Lv, Xin’e Shi, Gongshe Yang, Jianjun Jin

**Affiliations:** 1Laboratory of Animal Fat Deposition and Muscle Development, College of Animal Science and Technology, Northwest A&F University, Yangling 712100, China; 2Key Laboratory of Animal Genetics, Breeding and Reproduction of Shaanxi Province, College of Animal Science and Technology, Northwest A&F University, Yangling 712100, China

**Keywords:** CRISPR activation, CRISPR inhibition, histone modification, DNA methylation, transcription regulation

## Abstract

CRISPR/Cas9-mediated cleavage of DNA, which depends on the endonuclease activity of Cas9, has been widely used for gene editing due to its excellent programmability and specificity. However, the changes to the DNA sequence that are mediated by CRISPR/Cas9 affect the structures and stability of the genome, which may affect the accuracy of results. Mutations in the RuvC and HNH regions of the Cas9 protein lead to the inactivation of Cas9 into dCas9 with no endonuclease activity. Despite the loss of endonuclease activity, dCas9 can still bind the DNA strand using guide RNA. Recently, proteins with active/inhibitory effects have been linked to the end of the dCas9 protein to form fusion proteins with transcriptional active/inhibitory effects, named CRISPRa and CRISPRi, respectively. These CRISPR tools mediate the transcription activity of protein-coding and non-coding genes by regulating the chromosomal modification states of target gene promoters, enhancers, and other functional elements. Here, we highlight the epigenetic mechanisms and applications of the common CRISPR/dCas9 tools, by which we hope to provide a reference for future related gene regulation, gene function, high-throughput target gene screening, and disease treatment.

## 1. Introduction

### 1.1. Overview of CRISPR/Cas9

Gene editing technology is developing rapidly and is a feasible method for accurate and direct DNA editing. CRISPR repeats were discovered in *Escherichia coli* in 1987, and it was officially termed clustered regularly interspaced short palindromic repeats (CRISPR) in 2002 [1,2]. However, its DNA splicing function and role in the immune response of bacteria were only discovered in 2010. In 2012, Jennifer Doudna and Emmanuelle Charpentier clarified the biochemical nature of the CRISPR system, earning them the 2020 Nobel Prize in Chemistry [3,4,5]. Subsequently, several teams have modified the CRISPR/Cas9 system and applied it to eukaryotic cells [6,7]. Compared to zinc finger nucleases (ZFNs) and transcriptional activator-like effectors (TALENs), the CRISPR system has been more widely used in gene editing because of its high specificity, high efficiency, and easy reprogramming. Hitherto, the CRISPR system is divided into six types and twenty-one subtypes, among which the classic CRISPR/Cas9 system and newly discovered CRISPR/Cas12a and CRISPR/Cas13 systems have been applied in RNA imaging and rapid nucleic acid detection [8,9,10].

The CRISPR system is divided into two categories based on the system composition as the effector of the prokaryotic immune system. One CRISPR/Cas system category contains multiple effector proteins, including three types: types Ⅰ, Ⅲ, and Ⅳ. The other category comprises single multi-domain protein effectors widely used in current research and consists of types Ⅱ, Ⅴ, and Ⅵ. The widely used CRISPR/Cas9 system belongs to the type Ⅱ category, while the CRISPR/Cas12 and CRISPR/Cas13a systems belong to the types Ⅴ and Ⅵ categories, respectively. The CRISPR/Cas9 system has been widely applied in gene knock-in and knock-out, genome high-throughput screening, modeling, and other aspects of research owing to its relatively early discovery, our current better understanding, relatively simple structure, and ease of modification.

The CRISPR/Cas9 system mainly includes the Cas9 protein for cutting DNA and crRNA (CRISPR-derived RNA) and tracrRNA (trans-activating crRNA) for targeting. crRNA forms double-stranded RNA with tracrRNA through base complementary pairing and assembles into a complex with the Cas9 protein to target specific DNA sequences. Cas9 proteins recognize protospacer adjacent motif (PAM) sequences and exert endonuclease activity to cut DNA sites, resulting in DNA double-strand breaks (DSBs). Next, cells activate the DNA repair pathway to repair the damaged DNA strand. In the repair process, DNA is inserted, deleted, or reorganized, leading to modification of the target DNA sequences. Today, crRNA and tracrRNA are usually fused into a single RNA, termed sgRNA (single guide RNA), when using CRISPR tools to manipulate DNA molecules (Figure 1).

### 1.2. Application of CRISPR/Cas9

#### 1.2.1. Knock-Out or Knock-In of Target Genes Using CRISPR/Cas9 Technology

Knock-out or knock-in of DNA fragments by CRISPR/Cas9 mainly depends on Cas9 nuclease activity to produce DSBs and the subsequent initiation of cellular DNA strand self-repair. Generally, cells use efficient non-homologous end connection (NHEJ) to repair broken DNA, in which base insertion and deletion usually lead to frameshift mutations of the codon, resulting in inactivation of the target gene to achieve gene knock-out. In 2013, the CRISPR/Cas9 system was used for the first time to knock out single and multiple genes in cells [11,12]. In the same year, a multi-gene knock-out mouse model was rapidly and efficiently prepared by the CRISPR/Cas9 technique [13]. Over the last decade, CRISPR has become the most common tool for gene knock-out in cells and living animals [14,15]. For example, in exploring treatment for Duchenne muscular dystrophy (DMD), a muscle disorder caused by mutations in the DMD gene, researchers used CRISPR/Cas9 in patient-derived cells and mouse models to delete the exons from the mutated DMD gene to restore a normal reading frame to correct different mutations in DMD genes [16,17], which has been reviewed in more detail elsewhere [18,19]. Knock-out with a single sgRNA essentially mutates the nucleic acid sequence, affecting the protein it encodes. Compared with mRNA, non-coding RNA, including long non-coding RNA (lncRNA) and circular RNA (circRNA), cannot encode proteins. Thus, frameshift mutations or base deletions caused by base mutations do not affect the function of non-coding RNA. Therefore, two sgRNAs are designed to target both ends of the target non-coding RNA to knock out the non-coding RNA from the genome. Our previous research used CRISPR/Cas9 technology to delete all lncRNA *SYISL* and *lncMGPF* sequences to knock out the lncRNA. Besides being used to knock out genes, CRISPR/Cas9 can also knock in genes at a target site. After the DNA double-strand breaks, the broken part of the genome will undergo homologous recombination repair (HDR) according to the template (donor vector), thus realizing gene knock-in. Recently, CRISPR/Cas9 technology has been used to knock in genes at the cellular level in mice, zebrafish, livestock, and other animals. Gu Hao et al. (2021) used CRISPR/Cas9 technology to knock the PPAR-γ gene into the pig genome and breed PPAR-γ skeletal muscle-specific expression transgenic pigs, which increased oxidized muscle fiber content and intramuscular fat deposition [20]. This research has shown that as a gene editing innovation, CRISPR/Cas9 technology provides greater possibility for the precise manipulation of genes.

#### 1.2.2. High-Throughput Screening of Key Genes by CRISPR/Cas9

Besides knocking in or out DNA sequences, CRISPR/Cas9 technology can be used to conduct high-throughput screening of target genes in the whole genome [7,21]. Zhang Feng et al. [7] designed a genome-scale CRISPR/Cas9 knock-out (GeCKO) library containing 64,751 sgRNA targeting 18,080 genes. The lentivirus-encapsulated GeCKO library was transfected into melanoma cells, and new melanoma resistance-related genes NF2, CUL3, TADA2B, and TADA1 were identified by vemurafenib resistance screening. Besides high-throughput screening of functional genes, CRISPR/Cas9 is also used for high-throughput identification of important regulatory sequences, such as enhancer elements [22,23] and non-coding genomes [24,25]. Research has shown that CRISPR/Cas9-based genome screening technology could provide directional guidance to explore underlying key phenotypes of genes.

### 1.3. Construction of CRISPR/dCas9 Tools

The CRISPR tool allows for precise DNA editing, while the insertion or deletion of large DNA sequences into genomes can lead to structural changes that deteriorate their stability [26,27]. Therefore, constructing the CRISPR/Cas9 tool, which regulates gene expression at the transcriptional or post-transcriptional level without breaking the genome sequence, is considered a key approach to applying CRISPR technology. Hiroshi Nishimasu et al. [28] found that the Cas9 protein mainly relies on two active domains, RuvC and HNH, to cut DNA strands. Mutation of the RuvC and HNH domains of the Cas9 protein results in a loss of endonuclease activity and inability to cleave the DNA strands. However, it retains its ability to bind to the target DNA with the help of the gRNA. Thus, activation or inhibition domains can be linked to the dCas9 protein, and the targeting capabilities of the complex could be used to alter the chromosomal modification status of specific gene regulatory elements precisely (Figure 2). Several dCas9 tools that can activate or inhibit gene expression at the transcriptional level have been developed and successfully applied in gene expression regulation, gene screening, disease research, and the development of new therapeutic strategies. Here, we highlight and compare the mechanisms and application of common CRISPR/dCas9 tools to provide a reference for future research (Table 1).

## 2. CRISPRi Tools: Suppressing Gene Expression at the Transcriptional Level

### 2.1. dCas9-KRAB

Approximately 350 protein-coding genes in the human genome contain the Kruppel-associated boxes (KRAB) domain, which has transcriptional inhibition effects [37,38]. Gilbert et al. [6] fused the dCas9 protein with the KRAB domain to construct a dCas9-KRAB fusion protein, which could specifically target the EGPF vector to reduce fluorescence activity under the guidance of sgRNA. Further studies showed that dCas9-KRAB inhibits the expression of target genes by recruiting the methyltransferase SETDB1 to target sites. dCas9-KRAB can also target gene regulatory elements. For example, dCas9-KRAB can target HS2 enhancers, increasing enhancer H3K9me3 modifications, reducing the accessibility of enhancer and promoter chromosomes, and silencing the expression of multiple globin genes (Figure 3A,B) [29]. Although the platform fused the KRAB domain from KOX1 and could specifically suppress the expression of target genes in most cases, it cannot completely silence gene expression in some cases, which may affect subsequent research [39,40]. Alerasool et al. [41] tested the ability of silencing gene expression in 57 human KRAB domains and found that the potency of the KRAB domain significantly affected gene expression silencing. A comparison of the ability of proteins fused with different KRAB domains to inhibit gene expression showed that the KRAB domain from KOX1 is not a strong effector. By contrast, ZIM3 KRAB always inhibited gene expression effectively.

Currently, the dCas9-KRAB fusion protein is the most widely used tool for gene suppression, gene function research, gene regulatory element screening, and screening for disease treatments [31,42,43,44]. Liu et al. [45] used dCas9-KRAB fusion protein screening to profile globin promoters and identified an activating element near the BCL11A binding site, which coincided with the CCAAT box. CUT&RUN and base editing indicated that the NF-Y activator occupied the proximal CCAAT box, and the BCL11A initiated inhibition by binding competitively with NF-Y. Besides inhibiting gene expression, the dCas9-KRAB fusion protein can induce DNA methylation to explore the epigenome [46,47]. Thakore et al. [29] targeted dCas9-KRAB to the HS2 enhancer that realizes specific induction of H3K9 trimethylation, which silenced the expression of multiple globin genes. Additionally, dCas9-KARB has been used to develop novel strategies to treat liver cancer, alcoholic fatty liver disease, metastatic cancer, and lymphoma by regulating histone modification and gene transcription [47,48,49,50,51]. FoxP3 deletion leads to spontaneous breast cancer, and chromosome inactivation may lead to the inactivation of the normal FoxP3 gene in mice with heterozygous mutations. Cui et al. [52] found that targeting the reactivation of non-mutated alleles on the inactivated X chromosome (XCI) inhibited tumor growth while targeting inhibition of the XIST of XCI enhanced and prolonged the activation of FoxP3, providing a potential treatment for female breast cancer. Furthermore, dCas9-KRAB could also be used to develop methods to identify gene regulatory elements. Gasperini et al. [53] combined CRISPR and the expression of quantitative trait loci to propose crisprQTL mapping, a framework for the expression of multiple quantitative trait loci. The framework used dCas9-KRAB to interfere with each of the 5920 candidate enhancers in cells, followed by single-cell RNA sequencing. This study identified 644 cis-enhancer gene pairs using this framework and 471 enhancer-gene pairs with high confidence, suggesting that crisprQTL mapping will facilitate large-scale mapping of enhancer-gene regulatory interactions. 

### 2.2. dCas9-DNMT3A

DNA methylation transferases (DNMTs) can catalyze CpG island DNA methylation and inhibit gene expression [54,55,56]. Vojta et al. [30] linked the dCas9 protein to the catalytic domain of DNMT3A. They developed a dCas9-DNMT3A fusion protein, which caused direct DNA methylation in the promoter region of IL6ST and BACH genes, inhibiting gene expression (Figure 3C). Liu et al. [57] targeted methylation of CTCF rings using dCas9-DNMT3A fusion proteins, blocking CTCF binding and interfering with DNA cycling, which changed the expression of genes in adjacent rings. The dCas9-DNMT3A fusion protein could target the promoter regions of EpCAM, CXCR4, TFRC, and other genes and cause methylation of the target site CpG island, leading to transcriptional inhibition by fusion protein-introduced DNA methylation [58]. However, the off-target effects of the dCas9-DNMT3A fusion protein limit the evaluation of DNA methylation. To overcome this deficiency, Pflueger et al. [59] fused the SunTag array into the dCas9 protein to construct the dC9Sun-D3A system, which uses dCas9-SunTag to recruit proteins to the target site. This could independently regulate the expression of the DNMT3A catalytic domain and dCas9-SunTag and achieve more accurate and efficient DNA methylation editing. The dCas9-DNMT3A fusion protein has also been used to explore the mechanisms of disease and potential therapeutic targets. Tarjan et al. [60] found that dCas9-DNMT3A perturbed stable, partially heritable individual CTCF insulators, which were used to simulate the insulator loss mechanism associated with brain tumors. At the same time, Wu Jixiang et al. [61] used a dCas9-DNMT3A fusion protein to evaluate the effect of promoter methylation on SMARCA2 expression. They found that promoter methylation-induced SMARCA2 inactivation had carcinogenic effects in lung adenocarcinoma, which may be a potential target for clinical treatment. In 2021, Tiane et al. [62] targeted inhibition of the Id2 and Id4 DNA protein inhibitors by dCas9-DNMT3A and found that DNA methylation of Id2 and Id4 expression is important in the differentiation of oligodendrocyte progenitor cells, which is maladjusted in multiple sclerosis.

### 2.3. dCas9-HDAC

Histone deacetylase (HDAC) mainly catalyzes the process of deacetylation in vivo, which could inhibit target gene expression by reducing histone acetylation levels in the gene regulatory region [63,64]. Chen et al. [31] used a dCas9-HDAC8 fusion protein to block histone acetylation of the Fos gene enhancer (Figure 3D), thereby inhibiting the expression of Fos, which significantly reduced the ratio of neuronal ON, prolonged the time of the promoter in the OFF state, and reduced the overall frequency of burst. Activity-induced histone deacetylation can regulate gene transcription to affect neuronal gene expression and cell function. Like other CRISPR tools, dCas9-HDAC can induce genome deacetylation, which is used to investigate the impact of epigenetic modifications on biological processes. The pioneer factor PU.1 increased the acetylation of the DPP4 genomic region and promoted liver metastasis of colorectal cancer. Wang et al. [51] used a dcas9-HDAC fusion protein to target the DPP4 promoter to reduce its histone acetylation, inhibiting DPP4 expression and significantly reducing tumor growth and metastasis, suggesting a potential therapeutic strategy for chromatin remodeling in metastatic cancer.

### 2.4. CRISPRoff

The CRISPRi tool reportedly regulates targeted gene expression for a short time and is not heritable. Nunez et al. [32] fused DNMT3A, DNMT3L, and KRAB to the dCas9 protein to construct CRISPRoff structures (DNMT3A-DNMT3L-dCas9-KRAB) (Figure 3E,F). Transient CRISPRoff expression resulted in highly specific DNA methylation and gene repression maintained through cell division and differentiation of stem cells. Moreover, epigenetic silencing of CRISPRoff is not limited to genes with typical CpG islands and can silence genes without CpG island structures. Additionally, as it targeted many promoters and enhancers of different genes, CRISPRoff exhibited extensive and stable gene silencing in the genome, indicating that CRISPRoff can induce stable epigenetic memory. CRISPRoff provides more options to investigate the impact of genetic methylation modifications on gene function and flexible gene expression regulation than other CRISPRi tools.

**Figure 3 ijms-24-14865-f003:**
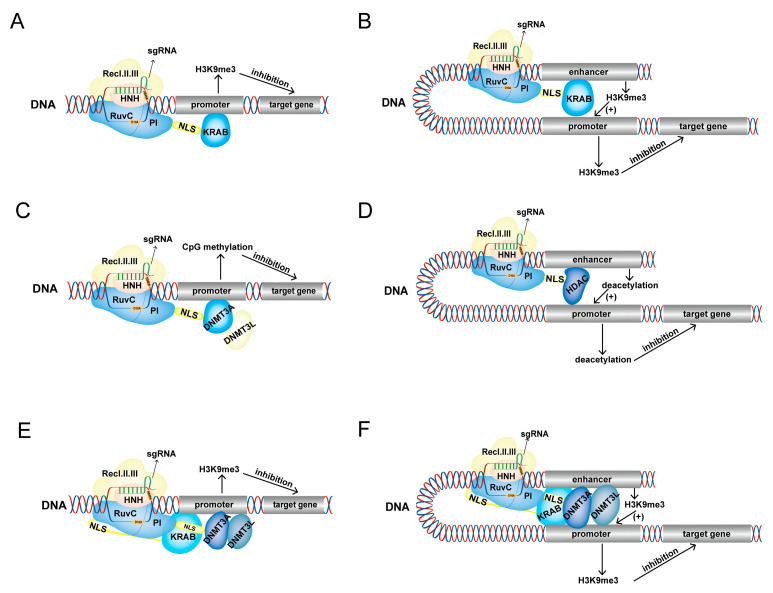
Inhibition of target gene expression with CRISPRi tools. (**A**,**B**) The dCas9 protein recruits the KRAB domain to the promoter (**A**) or enhancer (**B**) region of the target gene and catalyzes H3K9me3 of the promoter or enhancer, leading to inhibition of target gene expression. (**C**) dCas9 targets DNMT3A in the promoter region of the target gene and recruits the DNMT3L domain to catalyze CpG methylation, inhibiting the expression of the target gene. (**D**) dCas9 recruits the HDAC domain to the enhancer region of the target gene, activating the enhancer and resulting in promoter deacetylation and inhibition of target gene expression. (**E**,**F**) The CRISPRoff complex is directed by sgRNA to target the promoter (**E**) or enhancer (**F**) region of the target gene to catalyze the occurrence of H3K9me3 in the promoter and enhancer, inhibiting expression of the target gene.

## 3. CRISPRa Tools: Activating Gene Expression at the Transcriptional Level

### 3.1. dCas9-VP64

VP64 is a tetramer of herpes simplex virus 16 (VP16) that activates gene expression at the transcriptional level. Gilbert et al. [6] investigated the dCas9-VP64 fusion protein, effectively activating Gal4 UAS-GFP expression (Figure 4A). Furthermore, dCas9-VP64 significantly increased gene expression under the guidance of single or multiple sgRNA targeting NTF3 and VEGFA, indicating that this fusion protein could specifically activate the expression of endogenous human genes [33].

dCas9-VP64 is an effective CRISPR activation tool and has been widely used to study the activation of multiple genes. Saayman et al. used dCas9-VP64 to simultaneously activate 23 sites on the LTR promoter of the HIV-1 virus. They identified a new hotspot of activation in these sites, providing a promising functional therapy for HIV/AIDS [65]. Xu et al. improved dCas9-VP64 by concatenating multiple sgRNAs to an expression vector, enabling the simultaneous expression of multiple sgRNAs and exhibiting higher editing efficiency [66]. dCas9-VP64 also affects the treatment of muscle fibrosis, paralysis, cancer, and other diseases. For example, Kemaladewi et al. used the dCas9-VP64 fusion protein to upregulate the expression of Lama1 in the skeletal muscle and peripheral nerves of MDC1A mice, preventing or correcting muscle fibrosis and paralysis. [67,68]. 

### 3.2. dCas9-VPR

P65, a member of the NF-κB family, is a transcription factor with a Rel-like domain that activates the expression of target genes [69,70]. Gilbert et al. [6] constructed a dCas9-p65 fusion protein, confirming that it could effectively activate the target gene. To further improve the activation efficiency, Jiang et al. [71] combined dCas9-VP64 with the MS2-p65-HSF1 co-activation medium to construct the CRISPRa-SAM system (Figure 4C). This system activated the genes Mx2 and B4galnt2 in PK-15 cells, which increased the antiviral activity of PK-15 cells against PRV or H9N2 viruses, demonstrating the potential of the CRISPRa-SAM system to activate porcine genes and improve porcine antiviral activity. Huang et al. [72] constructed a stable HSd3b-dCas9-MPH-HFF cell line using the CRISPRa-SAM system to activate the expression of the Nr5a1, Gata4, and Dmrt1 (NGD) genes. Simultaneous targeted activation of endogenous NGD gene reprogrammed HFFs into functional Leydigg-like cells, demonstrating an innovative technique to treat male androgen deficiency. Meanwhile, the CRISPRa-SAM platform has also been used to explore the role of ubiquitin. To study the effect of excessive ubiquitin on cellular mechanisms, Han et al. [73] developed a system to upregulate a library by combining dCas9-VP64 with sgRNA containing MS2-p65-HSF1. The system could effectively and reversibly induce UBC upregulation under normal conditions, proving this platform’s value in studying the role of ubiquitin. 

Since the dCas9-p65 fusion protein contains a single p65 activation domain, it has a limited activation effect on target genes. To improve its effect, Lin et al. fused the highly active VP64-p65-RTA domain with the dCas9 protein to construct a dCas9-VPR activation system (Figure 4B) [34]. Analysis of the characteristics of this system studied in *Drosophila* cells and in vivo showed that dCas9-VPR could effectively activate the target and cause the dominant phenotype in vivo. The dCas9-VPR system has already been used to upregulate gene expression to inquire about the functions of genes. Guo et al. [74] designed a Dox-induced dCas9-VPR-CRISPR-ON system that could specifically regulate NANOG expression. Besides specific activating protein-coding genes, the dCas9-VPR system can also effectively target the expression of lncRNA. To explore the role of lncRNA DANCR in chondrogenesis and bone healing properties of adipose-derived stem cells in rats, Nuong Thi Kieu et al. [75] (2021) specifically upregulated the expression of endogenous lncRNA DANCR by dCas9-VPR. Activating the expression of DANCR significantly promoted the healing of rat skulls, providing a potential treatment strategy for improving skull healing.

### 3.3. dCas9-p300

The p300 protein is an important histone acetyltransferase, which can activate gene expression and participate in various biological processes [76,77]. Using dCas9-VP64 as a control, Hilton et al. [35] designed three fusion proteins: a p300 full-length sequence (dCas9^FLp300^), a p300 acetyltransferase (HAT) active core region (dCas9^p300-core (WT)^), and a p300 acetyltransferase nuclear (HAT) active core region mutant (dCas9^p300-core(D1399Y)^). The dCas9^p300-core (WT)^ fusion protein promoted histone acetylation of the target gene promoter and distal regulatory element and activated target gene expression more effectively and specifically (Figure 4D,E). dCas9^p300-core (WT)^ has better inherent ligand adjustability than other CRISPRa transcriptional activation tools, which lays the foundation for future spatiotemporal acetylation regulation of specific genomic sites [78]. 

dCas9-p300 has been widely used to screen and identify target gene regulatory elements due to its high efficiency and specific transcriptional activation ability [79]. Millions of hypothetical regulatory elements have been found in genome-wide association analyses, and their functional decoding remains a challenge. Klann et al. [80] used CRISPR epigenome editing technology to regulate the activity of regulatory elements in the proteogenomic environment for high-throughput screening of their functions. As an application example, a lentivirus sgRNA library could be designed to target DNase I hypersensitive sites around genes and combined with functional screening by dCas9-p300 to identify known and unrecognized regulatory elements in human cells, proving the ability of this technique to make regulatory elements annotate functions in the primordial chromosome environment. Like other CRISPR/dCas9 regulatory tools, dCas9-p300 could also help investigations of the effects of elements on genes. The regulatory mechanism of the CFTR gene in different tissues is diverse and complex and is achieved by multiple regulatory elements. However, how these regulatory elements regulate CFTR gene expression is not clear. Kababi et al. [81] selected 18 high-priority regions and targeted them with dCas9-p300 and dCas9-KRAB to evaluate their ability to regulate CFTR expression. dCas9-p300 media significantly increased CFTR mRNA expression when targeting the upstream 44 kb region of the transcriptional initiation site. The results also showed that using dCas9-p300 to increase the expression of CFTR may improve the efficacy of therapeutic regulators and aid in the discovery of new therapeutic interventions for treating cystic fibrosis. The dCas9-p300 platform is a powerful tool for studying acetylation modification in biological processes due to its high efficiency in inducing acetylation of the target site. Chen et al. [31] used dCas9-p300 or dCas9-HDAC8 fusion proteins to simulate or block activity-induced acetylation at Fos gene enhancers to explore how histone acetylation regulates Fos gene transcription through transient and rapid changes. Increased histone acetylation prolonged the Fos gene outbreak time, increasing its transcription and eventually increasing Fos protein levels. 

### 3.4. dCas9-dMSK1

Mitogen and stress-activated protein kinase 1 (MSK1) catalyzes the phosphorylation of histones H3S10 and H3S28 [82,83]. Li et al. [36] designed three fusion proteins using the acetylation activity of MSK1 in catalytic phosphorylation: dCas9-MSK1, which connects the full length of MSK1, dCas9-dMSK1, which connects the catalytic core domain of MSK1, and dCas9-ddMSK1, which includes a mutation of dMSK1. The dCas9-dMSK1 fusion protein had better catalytic activity, resulting in improved histone H3S28 phosphorylation at the target site (Figure 4F). ChIP-seq and RNA-seq showed that dCas9-MSK1 could specifically bind and catalyze the phosphorylation of histone H3S28 on the promoter of the OCT4 gene and effectively induce its expression. This study showed that dCas9-dMSK1 is a powerful tool for studying histone phosphorylation and targeted activation genes, expanding the functions of this epigenetic editing tool.

**Figure 4 ijms-24-14865-f004:**
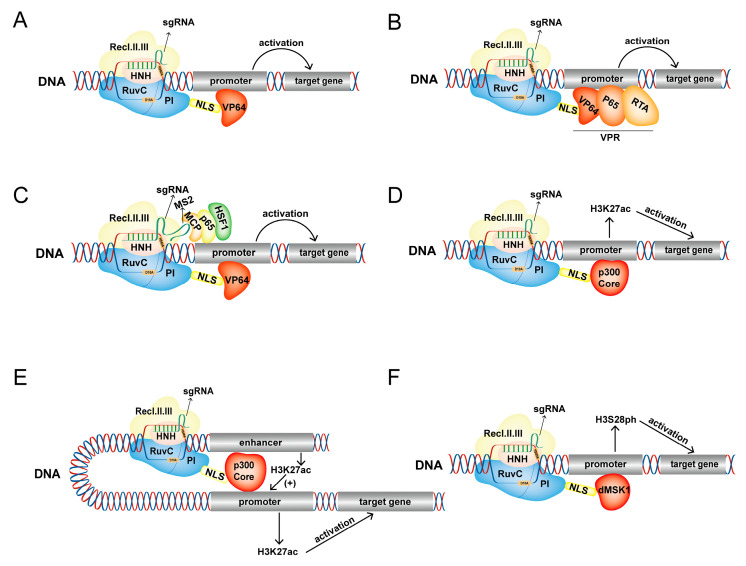
Activation of target gene expression by CRISPRa tools. (**A**,**B**) The dCas9 protein recruits the VP64 domain (**A**) or VPR activation system (**B**) to the promoter region of the target gene and acts on the promoter to activate the expression of the target gene. (**C**) MCP-p65-HSF1 complex recruited by MS2 co-activates the target gene expression with VP64 effector recruited by dCas9 protein. (**D**,**E**) dCas9 recruits the p300 core domain to the promoter or enhancer of the target gene to catalyze H3K27ac in this region, resulting in activation of gene expression. (**F**) dCas9 recruits the dMSK1 domain to the promoter of the target gene to catalyze the production of H3S28ph in this region, inducing activation of target gene expression.

## 4. Limitations of the CRISPR/dCas9 System

Although the CRISPR/dCas9 system is a specific and efficient tool in gene regulation and the induction of epigenetic modifications, its efficiency has some limitations, including off-target effects, cytotoxicity, vector delivery, and inhibition and activation. Many studies have shown that the CRISPR/dCas9 regulation platform is highly specific, and some CRISPR/dCas9 tools are rarely off-target at the transcription level, which is superior to the CRISPR/Cas9 platform [29]. However, off-target effects and cytotoxicity remain major hurdles in adopting the CRISPR/dCas9 regulation platform [84]. Recently, studies have focused on the off-target characteristics and strategies to improve specificity [59,85,86]. Another issue affecting the application of the dCas9 system is the delivery of expression vectors for dCas9-sgRNA into cells. The common delivery systems are divided into biological, chemical, and physical delivery. Dual-AAV-mediated vector delivery is the most practical among the existing delivery systems due to its efficiency [87]. However, limitations of AAV load, potential risks, inhibition of pre-existing anti-AAV antibodies, and immunogenicity mean that delivery remains a major challenge in applying CRISPR/dCas9 tools. Novel delivery systems are being explored to deliver expression vectors more safely and efficiently. Currently, several novel and superior delivery methods, such as virus-like particle (VLP) delivery technology and multistage delivery nanoparticles, have been discovered, bringing an increased possibility of the CRISPR system being applied to gene therapy [88,89]. Targeted reviews of expression vector delivery systems have been summarized and evaluated in more detail, providing a high-value reference to improve delivery methods and promote application [90,91]. In addition, the inhibition and activation efficiency also impact the function of the CRISPR/dCas9 system. Fusion protein-linked effectors are critical to the efficiency of the platform. Recently, more powerful effectors or systems have been investigated, and these improvements will make the CRISPR/dCas9 system more widely applicable [41,92].

## 5. Conclusions and Prospects

The discovery and application of CRISPR/Cas9 technology provide broader prospects for gene editing technology and greater possibilities for more precise manipulation of DNA. The recognition method based on base complementary pairing makes the CRISPR/Cas9 technology highly specific and easy to reprogram, making it safer and more efficient to achieve DNA knock-out and knock-in. However, the genome is a spatial three-dimensional structure formed by the interaction of different components rather than a simple linear structure; specific DNA fragments may play important roles in forming this three-dimensional structure [93,94,95]. Thus, knock-out or knock-in of large DNA fragments may affect the formation of the three-dimensional structure of the genome, with unpredictable risks. The emergence of CRISPR/dCas9 technology provides a feasible solution to this problem. Compared with CRISPR/Cas9, CRISPR/dCas9 works mainly by linking effector proteins, and activating or silencing gene expression is primarily modified by epigenetic modification without affecting the genome sequence, avoiding unpredictable effects and minimizing toxicity to cells. 

Due to the higher safety of CRISPR/dCas9 gene regulation tools, the CRISPR/dCas9 tools have been used to change epigenetic markers of the genome for disease treatment, providing a potential epigenetic therapeutic strategy to treat complicated diseases [51,84]. Although CRISPR/dCas9 is commonly used to regulate gene expression and induce epigenetic modification, playing an important role in gene activation, silencing, identification of gene functional elements, exploration of the epigenome, and high-throughput screening, CRISPR/dCas9 technology is less frequently used in gene therapy due to the delivery and off-target risk [96,97,98,99]. As researchers seek the means to improve delivery and reduce off-target effects [100,101,102], coupled with its small impact on the genome and superior nucleic acid localization ability, CRISPR/dCas9 will have great potential in gene therapy, bringing hope for the clinical treatment of more diseases. 

Additionally, the ability to accurately target DNA sequences with the dCas9 protein complex may be combined with other methods to improve current gene research techniques and become a powerful tool for understanding gene expression and regulation in cells. Zhang et al. [103] proposed a paired dCas9 (PC) in vitro detection report system by combining the localization ability of the dCas9 protein complex with a mitotic luciferase reporter. The PC detection system showed good stability and sensitivity in detecting the genome of *Mycobacterium tuberculosis* (Mtb). Guk et al. [104] combined CRISPR/dCas9 with SYBR Green I (SG I) and used dCas9/sgRNA-SG I to realize rapid and sensitive detection of methicillin-resistant *Staphylococcus aureus*. This method is simple, highly efficient, and sensitive, ensuring that the dCas/sgRNA-SG I detection system can be applied to detect various pathogens. The excellent localization ability of the dCas9 protein complex can be used for live-cell DNA imaging. In a study of DNA–protein and DNA–RNA–protein interactions, new techniques were developed based on the specific binding ability of the dCas9 protein complex to DNA–protein and DNA–RNA–protein complexes. For example, dCas9-gRNA-guided chromatin immunoprecipitation (dCas9-ChIP) was used to study the interaction of the Vig and Vig2 transcription factors with histone H2A and lncRNA HUMT- and transcription factor YBX1-mediated regulation of FOXK1 gene transcription [105,106]. Wang Yucheng et al. [107] identified the upstream regulatory factor of the BpNAC090 transcription factor in *Betulaphylla* using reverse chromatin immunoprecipitation (R-ChIP-dCas9), which is based on the CRISPR/dCas9 system. 

Based on the above, the CRISPR/dCas9 system provides the possibility to regulate gene transcription without changing the genome, which reduces the toxicity to cells. Superior nucleic acid positioning ability, easy-to-modify structures, and powerful regulatory capabilities enable the CRISPR/dCas9 platform to more likely correct gene expression. All of these make safe gene therapy more possible to offer more strategies for difficult diseases, but the delivery and off-target effects remain a major challenge for clinical. The limitation and immunogenicity of delivery will prevent the protein from the cells, while the risk of off-target effects may erroneously affect gene transcription. Thus, both efficient and safe delivery methods and reliable off-target assessment methods need to be focused on to improve CRISPR/dCas9 tools. As these problems are gradually overcome, CRISPR tools will be safe enough to be used in clinical gene therapy in the future.

## Figures and Tables

**Figure 1 ijms-24-14865-f001:**
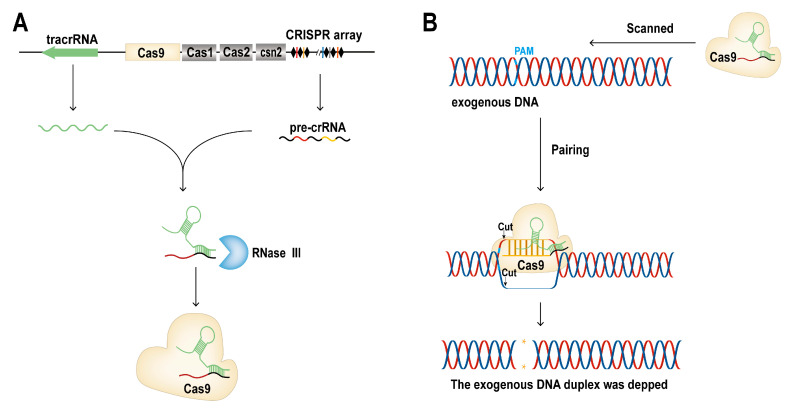
Schematic diagram of the CRISPR/Cas9 working principle. (**A**) CRISPR gene is transcribed to form tracrRNA and crRNA, combined by base complementary pairing, and matured under the action of RNase III. Next, they are combined with Cas9 to form the Cas9 complex for DNA cutting. (**B**) The Cas9 protein complex scans the PAM sequence of exogenous DNA and combines with the target sequence according to base complementary pairing. The Cas9 protein cuts the DNA double strand by endonuclease activity, resulting in double-strand breakage of the target DNA.

**Figure 2 ijms-24-14865-f002:**
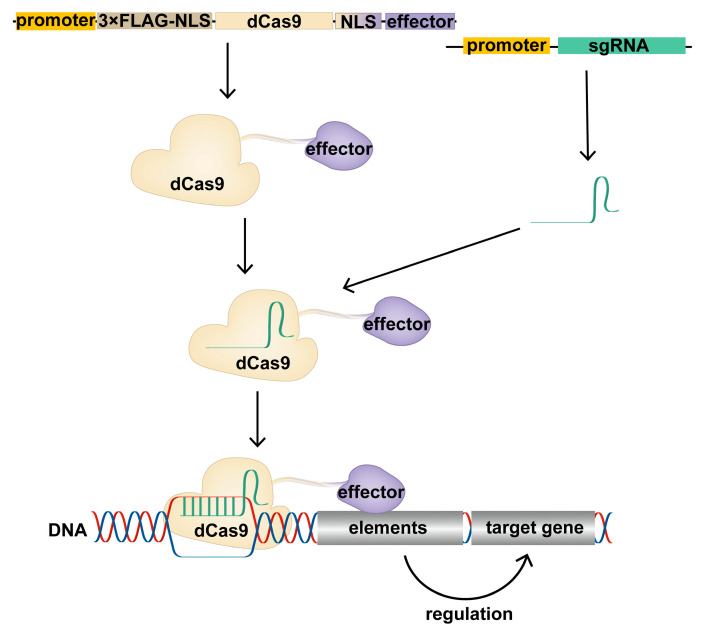
Schematic of the CRISPR/dCas9 regulatory target gene. The expression vector expresses the dCas9 fusion protein in cells, which binds the transcribed sgRNA to form the CRISPR/dCas9 regulatory tool, resulting in the recruitment of the effector domain to the promoter or enhancer region of the target gene under the guidance of sgRNA. Effector domains act on promoters or enhancers of target genes to modify these regions, regulating target gene expression.

**Table 1 ijms-24-14865-t001:** Functional elements of CRISPR-dCas9 tools and epigenetic principles of regulation expression.

	Tools	Element	Effector	Epigenetic	Gene Expression	Reference
CRISPRi	dCas9-KRAB	promoter, enhancer	KRAB	H3K9me3	downgrade	[6,29]
	dCas9-DNMT3A	promoter	DNMT3A	CpG methylation	downgrade	[30]
	dCas9-HDAC	enhancer	HDAC	deacetylation	downgrade	[31]
	CRISPRoff	promoter, enhancer	KRAB, DNMT3A, DNMT3L	H3K9me3	downgrade	[32]
CRISPRa	dCas9-VP64	promoter	VP64	-	upregulate	[6,33]
	dCas9-VPR	promoter	VPR	-	upregulate	[6,34]
	dCas9-p300	promoter, enhancer	p300	H3K27ac	upregulate	[35]
	dCas9-dMSK1	promoter	dMSK1	H3S28ph	upregulate	[36]

Abbreviations: CRISPRi: CRISPR inhibition; CRISPRa: CRISPR activation; H3K9me3: Histone 3 trimethylated on lysine 9; H3K27ac: Histone 3 acetylation on lysine 27; H3S28ph: Histone 3 phosphorylation on serine 28.

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
