# Peer review of "CRISPR/dCas9 Tools: Epigenetic Mechanism and Application in Gene Transcriptional Regulation"

_ijms, 2023, doi:10.3390/ijms241914865_

Round 1

Reviewer 1 Report

Manuscript ID: ijms-2612642

Title: CRISPR/dCas9 Tools: Epigenetic Mechanism and Application in Gene Expression Regulation

The clustered regularly interspaced short palindromic repeats (CRISPR)/Cas9 system, the newly developed CRISPR/dCas9, has been repurposed for transcriptional regulation. This system consists of three main components: a catalytically inactive Cas9 protein (dCas9), a customizable single guide RNA (sgRNA) complementary to the promoter region of a gene, and a transcriptional effector, either transcriptional activators (CRISPRa) or repressors (CRISPRi). Transcriptional interference is achieved by blocking RNA polymerase binding or elongation upon binding of the dCas9/sgRNA and transcriptional effector complex to the promoter region of the downstream target gene. The CRISPR-Cas9 system has the advantages of simplicity, high efficiency, high specificity, and low production cost. It greatly facilitates the study of gene function. CRISPRi works in an analogous manner to RNAi. Both systems aim to silence or knock down gene expression, but have different mechanisms and principles. In essence, CRISPRi silences gene expression at the DNA level by preventing transcription, whereas RNAi uses a post-transcriptional mechanism by cleaving transcribed mRNAs.

This manuscript describes the CRISPR/dCas9 Tools: Epigenetic Mechanism and Application in Gene Expression Regulation. Overall, the topic discussed is interesting, but the manuscript needs some improvements that, in my opinion, will make the manuscript more consistent for publication.

-Abstract

The abstract needs to be revised, as it needs to start with a short general statement/your aim (topics) and end with a short conclusion with a related suggestion, if any.

-Introduction:

The introduction must be a separate section (1-Introduction), and after a short introduction to CRISPR and CRISPR/dCas9, the advantages/disadvantages, differences to other systems, gaps in this context, as well as briefly previous work should be presented. Make sure you explain the need for the review and what you are trying to accomplish.

- Change "Discussion and Prospects" to "Conclusion and Prospects" if not, this section needs to be more in-depth and discuss all of the topics that have been covered.

Minor editing of English language required

Reviewer 2 Report

The review focuses on CRISPR/dCas9 tool's application for gene regulation. The review is well written and comprehensive. I have minor suggestions for authors:

a) Currently, the figures are oversimplified version of what is already available in existing reviews and papers. Most of the researchers are aready familiar with CRISPR mechanism. I will suggest authors to include figures that are more informative and covers latest trend in CRISPR technology.

b) The information on applications of CRISPR/dCas9 technology can be covered in tabular format to organize the information for better comparison.

c) There are too many references, the authors might consider to use latest reviews as a common reference to avoid referencing several papers individually.

Reviewer 3 Report

The present manuscript is a very systematic and interesting review of CRISPR/dCas techniques, their function (mechanism) and their application. Beginning with their discovery, the mechanisms of activity are explained and very well illustrated. Their application in gene modification (knock-in and knock-out) and high-throughput screening for relevant genes is described with several eminent examples in the field, and novel references are included. Further, the functional elements of the system are explained, as well as the mechanism behind their function, and inhibitory and activating CRISPR tools are listed, again with several examples. The limitations of the system are summarized, and the review completes with an insight into future perspectives of the methodology and less known options for applications.

This is a comprehensive review which I am sure will attract several readers, beyond those using CRISPR/dCas already. I would recommend the authors to (1) point the readers to Figures and Tables already in the article text and (2) summarize the article in a Conclusions paragraph. Please find below a list of remarks, which I hope you will find helpful.

Line 136: Title to Table 1: please include the explanation of abbreviations CRISPRi and CRISPRa. Table 1 is also not cited in the text, and the “H3K9me3, H3K27ac and H3S28ph” could be explained alongside.

Line 288: Drosophila in italics

Line 297: “promoted the healing of 297 rat skulls, providing a potential treatment for improving skull healing.” – perhaps providing a potential treatment strategy?

Line 369-370: inhibition of pre-existing anti-AAV antibodies could also be listed

Line 397: “Due to the higher safety of CRISPR/dCas9 gene regulation tools are safer”- please reword

Line 423: “Betulaphylla“ – please the species name in italics

From line 425: these sentences are optimally suited for a “Conclusion” paragraph, which could also summarize all potential novel applications of the method described (e.g. for highly sensitive detection and identification of contaminating and pathogenic microorganisms, identification of regulatory factors involved in transcription, etc.

Reviewer 4 Report

This manuscript entitled "CRISPR/dCas9 Tools: Epigenetic Mechanism and Application in Gene Expression Regulation" by Cai R. et al. summarized Gene Expression Regulation with CRISPR/dCas9 Tools. The manuscript's main strength is that it addresses a timely and fascinating topic, providing a comprehensive review of CRISPR/dCas9 and and discussing the latest research. I think the idea of this article is really interesting, and the authors’ fascinating observations on this timely topic may be of interest to the readers. However, some comments, as well as some crucial evidence that should be included to support the author’s argumentation, needed to be addressed to improve the quality of the manuscript, its adequacy, and its readability prior to its publication in the present form. 

Title: This is the most important section of the manuscript. Please present a concise and self-explanatory title stating the most important message of this review.

Abstract: I suggest the authors present the background, a short summary, and a conclusion. The general background (one to two sentences), the specific background (two to three sentences), and the current issue covered by this review (one sentence) should all be included in the background. I would like the author to provide background information, a problem statement. The conclusion should begin with one sentence that summarizes the main message using words like "Here we highlight." The authors should describe the potential and the advancement this study has made in the field in the first sentence of the conclusion, followed by two to three sentences that provide a broader perspective that is easily understood by a scientist.

A graphical abstract that will visually summarize the main findings of the manuscript is highly recommended.

Introduction: I would like the authors to reorganize this section with several paragraphs, introduce information on the key study constructs that should be understood by readers, and make it persuasive enough to advance the primary goal of the author's recent research and the particular goal the author has intended by this review. I would like to suggest that the authors present the introduction beginning with the overall context, and concluding with the current problem addressed in this review. I also recommend that the authors provide the rationale for presenting subsequent sections in order to assist the reader. 

 DiscussionI expect the authors to develop arguments clarifying the potential of this study as an extension of the previous work, the implication of the findings of this study, how this study could facilitate future research, the ultimate goal, the challenge, the knowledge and technology necessary to achieve this goal, the statement about this field in general, and finally the importance of this line of research. It is particularly important to present the limits, merit, and potential translation of this study to clinical practice.

it may need moderate corrections.
